

# Effects of vegetation restoration in karst areas on soil nitrogen mineralisation

Jianghong Wu, Xianghuan Gong and Yingge Shu

College of Agronomy, Guizhou University, Guiyang, Guizhou, China

Corresponding author
Yingge Shu, maogen958@163.com

## ABSTRACT

**Background:** Nitrogen mineralization plays a critical role in the ecosystem cycle, significantly influencing both the ecosystem function and the nitrogen biogeochemical cycle. Therefore, it is essential to investigate the evolutionary characteristics of soil nitrogen mineralization during the karst vegetation restoration to better understand its importance in the terrestrial nitrogen cycle.

**Methods:** This study analyzed from various stages of vegetation growth, including a 40-year-old woodland, 20-year-old shrubland, 15-year-old shrubland, 5-year-old grassland, and nearby cropland. The aerobic incubation technique was used for 35 days to evaluate soil N mineralization characteristics and their correlation with soil environmental factors. The study focused on examining the variations in soil N mineralization rate (NMR), N nitrification rate (NR), net nitrification rate (AR), and $NH_4^+$-N and $NO_3^-$-N levels.

**Results:** Nitrate nitrogen, the primary form of inorganic nitrogen, increased by 19.38% in the 0–40 cm soil layer of the 20-year-old shrubland compared to the cultivated land. Soil $NH_4^+$-N levels varied during the incubation period, decreasing by the 14th day and rising again by the 21st day. Soil $NO^{3-}$-N and total inorganic nitrogen levels initially increased, then declined, and eventually stabilized, reaching their highest levels on the 14th day. During vegetation restoration, the soil NR and NMR decreased gradually with increasing incubation time. The 15-year shrub, 20-year shrub, and 40-year woodland showed the potential to increase soil NR and NMR. Furthermore, the 15-year shrub and 20-year shrub also increased soil AR. The Mantel test analysis indicated positive correlations among total nitrogen (TN), total phosphorus (TP), total potassium (TK), silicon (Si), AR, NR, and NMR. While available phosphorus (AP) and NMR demonstrated positive correlations with NR and NMR. Furthermore, TN, TP, TK, and Si were found to be positively correlated with AR, NR, and NMR, whereas AP and $NO_3^-$-N showed negative correlations with AR, NR, and NMR. It is worth noting that $NH_4^+$-N had the greatest effect on AR, while the bulk density (BD) significantly affected the NR. Furthermore, ammonium nitrogen (AN) and soil organic carbon (SOC) were identified as the primary contributors to NMR. This study provides a theoretical basis for comprehending the influence of vegetation restoration on soil nitrogen mineralization and its role in ecosystem restoration.

## INTRODUCTION

Nitrogen is crucial for the synthesis of a wide range of complex organic compounds and is essential for the survival of all living organisms on Earth (*McNeill & Unkovich, 2007*; *Pandey, Panda & Singh, 2024*). Soil comprises about 95% of the total nitrogen in terrestrial ecosystems around the globe, and soil nitrogen is considered an essential element that significantly influences plant growth, regulates nutrient cycling, and supports ecosystem functioning (*Knops, Bradley & Wedin, 2002*; *Zhang et al., 2014*; *Yadav et al., 2021*). Although the majority of nitrogen in the soil is found in organic forms, the nitrogen mineralization rate (NMR), which measures the ratio of organic to inorganic nitrogen, is a key indicator of how efficiently soil nitrogen is converted to a form available for plant uptake (*Risch et al., 2020*). The process of converting organic nitrogen to inorganic nitrogen, referred to as nitrogen mineralization, directly affects the availability of nitrogen in the soil (*Zhong & Makeschin, 2003*). This process is critical for plant growth and supports various ecosystem services. Soil microorganisms are responsible for the processes of ammonification and nitrification, which are essential to the mineralization of organic nitrogen (*Maslov & Maslova, 2022*). Several factors can influence nitrogen mineralization in the soil, including land use, soil characteristics, pH levels, temperature, moisture levels, type of vegetation, apoplastic quality, microbial populations, and human activities (*Templer et al., 2005*; *Hu et al., 2019*; *Risch et al., 2019*; *Maslov & Maslova, 2022*; *Zhang et al., 2022*; *Li et al., 2023*). Changes in nitrogen mineralization affect nitrogen availability (*Schlesinger & Bernhardt, 2013*), as well as primary productivity, ecosystem functioning, and long-term sustainability (*Chen, Zeng & Fahey, 2009*; *Heitkamp et al., 2008*). Furthermore, nitrogen cycling and transformation processes are significantly influenced by the microbial population in the soil and the soil carbon-to-nitrogen (C:N) ratios (*Chen et al., 2019*; *Padalia et al., 2022*; *Pandey et al., 2024*). Therefore, the characteristics of nitrogen mineralization serve as critical indicators for the assessment of soil quality.

Land use and restoration techniques have been shown to substantially influence soil nitrogen (N) mineralization in numerous studies (*Gurlevik & Karatepe, 2016*; *Li et al., 2018*; *Wang et al., 2017*). These effects are primarily due to variations in plant diversity and abundance, as well as variations in soil physical, chemical, and microbial properties under different restoration techniques (*Deng et al., 2014*; *Rhoades & Coleman, 1999*). The impact of vegetation on N mineralization depends on the vegetation type, which influences both the quantity and quality of organic matter and the efficiency of nitrogen uptake by plants (*Rahman, Bárcena & Vesterdal, 2017*; *Unver, küçük & Tufekcioglu, 2014*). Various research focused on the impact of land use changes on N transformations (*Contosta, Frey & Cooper, 2011*; *Li et al., 2014*). However, there is a debate about how N mineralization rates are affected by vegetation restoration (*Li et al., 2014*). Some studies reported an increase in N mineralization (*Gurlevik & Karatepe, 2016*; *Wang et al., 2017*), while others observed a decrease (*Li et al., 2014*; *Yang et al., 2010*), or found no significant change (*Zeng et al., 2009*). *Owen et al. (2003)* observed higher soil mineralization rates in forests compared to grasslands attributing to the variations in carbon assimilation among plant functional

groups and differences in soil characteristics. *Wei et al. (2017)* observed that, despite an increase in functional group abundance, root nitrogen content decreased together with higher biomass, resulting in a reduced net soil nitrogen mineralization rate. There are substantial seasonal variations in the availability and turnover of soil nitrogen (*Dujardin et al., 2012*). Microbial activity is directly influenced by environmental factors, including temperature, moisture, and pH (*Unver, küçük & Tufekcioglu, 2014*; *Ye et al., 2015*). *Dujardin et al. (2012)* observed that soil ammonium content reaches its highest level during the summer, due to increased microbial activity. *Hu et al. (2015)* observed similar soil nitrogen transformations in both biocrust-covered soils and bare ground, attributed to decreased microbial abundance and activity in extremely low temperatures. Despite the importance of soil nitrogen mineralization, there is limited research focusing on profiled soils in karst ecosystems. Furthermore, ecosystem responses to influencing factors can vary significantly (*Booth, Stark & Rastetter, 2005*; *Tapia-Torres et al., 2015*; *Zhou et al., 2009*), highlighting the necessity for site-specific assessments of nitrogen transformation (*Burke, 1989*; *Liu et al., 2017*). Moreover, the majority of ecosystems lack a comprehensive understanding of the impact of vegetation restoration on soil nitrogen mineralization and the influence of soil environmental factors on this process. These knowledge gaps limit the precise prediction of nitrogen biogeochemical cycling.

The Southwest Karst region is recognized as one of the world's three major continuous karst distribution areas (*Sheng et al., 2018*). Throughout the latter half of the 20[th] century, significant carbonate development in this region resulted in shallow soil layers, complex karst ecosystems, high population density, and frequent human activities. These elements contributed to significant vegetation loss and ecosystem degradation (*Wang, Liu & Zhang, 2004*). In response, the Chinese government has initiated various vegetation restoration projects in the region (*Basile-Doelsch, Balesdent & Pellerin, 2020*; *Chen et al., 2018*; *Wang et al., 2018*). Despite the increased vegetation cover achieved through these initiatives, the impacts (*Li et al., 2019*) and the underlying mechanisms of long-term restorations on soil inorganic nitrogen accumulation (*Li et al., 2019*; *Liu et al., 2024*) and nitrogen mineralization remain unclear.

This study analyzed ecosystems at various stages of natural succession following the retirement of agricultural land. The focus was on grasslands retired for 5 and 15 years, shrublands abandoned for 20 years, and woodlands left fallow for 40 years, with comparisons made to actively cultivated arable land as a baseline. This study was based on the hypothesis that vegetation restoration has a major effect on soil N mineralization. This effect was attributed to the continuous build-up of soil organic matter and significant changes in soil environmental factors. To validate this premise, the study aimed to achieve two primary objectives: (1) elucidate the mechanisms through which vegetation restoration affects soil N mineralization and (2) quantify the key soil physico-chemical parameters that influence this process. The primary objective was to verify the effects of vegetation restoration on soil N mineralization, assess the impact of key soil properties, and establish a strong scientific basis to guide ecological rehabilitation and soil management practices.
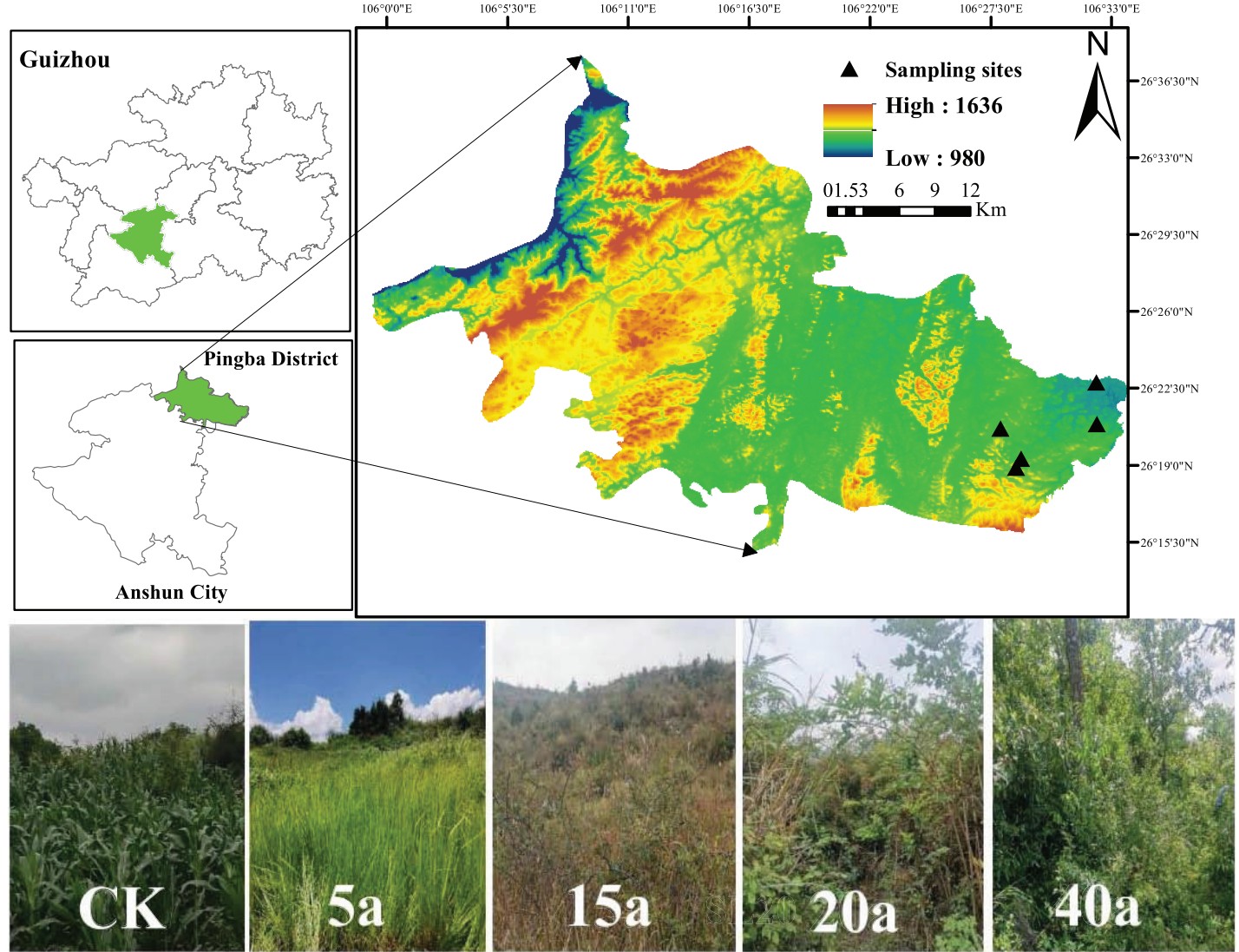

**Figure 1** **Basic information about the samples.** Sampling sites. Map data © 2024 Bigmap. 

## MATERIALS AND METHODS

### Study area

Pingba District (26°15′–26°37′40″N, 105°59′20″–106°33′43″E), Anshun City, Guizhou Province, lies within subtropical humid monsoon climate zone. The area, with an elevation ranging from 963 to 1,645.6 m and an average annual temperature of 13.3 °C, features complex topography characterized by typical karst landscapes and diverse restored vegetation types. The parent rock of the study area is dominated by limestone, whereas the soil is primarily calcareous (Fig. 1).

### Selection of sample plots

The vegetation types and restoration timelines of the area were accurately determined through detailed analysis using Google Historical Image Maps and on-site field surveys.

**Table 1 Basic information of the sample site.**

| Sample type | Recovery years/a | Altitude /m | Longitude and latitude | Predominant species |
|---|---|---|---|---|
| Cropland | 0 | 1,211 | 26°20′52″N, 106°32′18″E | *Zea mays* |
| Grassland | 5 | 1,217 | 26°20′59″N, 106°32′18″E | *Imperata cylindrical* |
| | 15 | 1,285 | 26°20′5″N, 106°27′56″E | *Imperata cylindrical* |
| Shrubland | 20 | 1,289 | 26°18′54″N, 106°28′39″E | *Pyracantha fortuneana, Artemisia annua, Rubus idaeus* L., *Zanthoxylum simulans* |
| Woodland | 40 | 1,223 | 26°19′19″N, 106°29′6″E | *CatalpabungeiC.A.Mey Celtis sinensis Pers* |

Afterward, a series of carefully chosen sample plots were identified, ensuring minimal variation in crucial factors like the type of restored vegetation, the duration of restoration, the topography, and the soil type. The selected restored vegetation types included grassland, shrub, shrub grassland, and woodland, corresponding to restoration periods of 5, 15, 20, and 40a, respectively. Conversely, adjacent cultivated land was used as a control (CK). The dominant vegetation type in grassland was Leucaena [*Imperata cylindrical (L.) Beauv*]. In shrub areas, key species included pyracantha (*Pyracantha fortuneana*), artemisia (*Artemisia annua*), wild berry (*Rubus idaeus* L.), and wild peppercorn (*Zanthoxylum simulans*). The woodland primarily featured Park and Rowan (*CatalpabungeiC.A.Mey, Celtis sinensis Pers*). While the cultivated land was predominantly planted with maize (*Zea mays*). Further details on the sample plots can be found in Table 1.

## Soil sampling

In July 2022, soil samples were collected from the study area following a precise protocol. The surface was first cleared of any debris and humus to maintain sample integrity. Using the "S" sampling method, five different soil horizons (0–5, 5–10, 10–20, 20–30, and 30–40 cm) were obtained from three selected soil profiles. To maintain soil structural integrity during transportation, samples were carefully sealed and laid flat. A total of 75 soil samples were collected and analyzed for their physicochemical properties through detailed laboratory assessments.

## Sample analysis and methods

Soil pH was determined using the potentiometric method with a water-to-soil ratio of 2.5:1. Total phosphorus (TP) and total potassium (TK) were analyzed through NaOH dissolution while AN was determined using the alkali diffusion method. Available phosphorus (AP) was determined by the 0. 5 mol·L$^{-1}$NaHCO$_3$ method and available potassium (AK) was assessed by ammonium acetate leaching flame photometry. Soil moisture content (SMC) was quantified through dehydration, soil bulk density (BD), and total portfolio porosity (STP) was determined by the ring knife method. Soil texture was

analyzed using the hydrometer method. The soil particles were classified according to the international system (*Ge et al., 2019*) (sand (Sa) 2–0.02 mm, silt (Si) 0.02–0.002 mm, and clay (Cl) < 0.002 mm). The methodology for specific references to the above indicators was based on guidelines from (*Sparks et al., 1996*).

The soil's $NH_4^+$-N was obtained using a 2 mol·L$^{-1}$ KCl solution and the indophenol blue colorimetric technique (*Lu, 1999*). On the other hand, the NO$^-$-N levels were determined through the dual-wavelength ultraviolet spectrophotometric approach along with a correction factor (*Norman, Edberg & Stucki, 1985*). $NH_4^+$-N and $NO_3^-$-N in soil samples were conducted utilizing the METASH UV-5500 UV-Vis spectrophotometer, a precision instrument sourced from METASH Instruments in Shanghai, China.

### Determination of mineralizable nitrogen

Soil organic nitrogen mineralization was assessed through aerobic incubation (*Stanford & Smith, 1972*; *Chenxiao et al., 2024*). Initially, 60 g of soil, sieved through a 2 mm mesh, were placed in 250 mL of PE clinker bottle for the incubation process. The soil's moisture content was then adjusted to 30% of its field water-holding capacity was set at 25 °C before being placed in a temperature-controlled incubator (ROX-250B). A 7-day pre-incubation period was carried out to restore soil microbial activity. After the pre-incubation stage, the samples were sealed with black cling film, which was punctured to allow for aeration, and then stored in darkness at 25 °C for 35 days. Aeration was performed every 3 days for 30 min, while moisture levels were carefully monitored by regularly weighing the sample. Destructive sampling was conducted on days 7, 14, 21, 28, and 35 following incubation. On each occasion, 10 g of soil were combined with 35 mL of 2 mol·L$^{-1}$ KCl solution (in a 5:1 ratio), shaken for 1 h and filtered into plastic containers for subsequent analysis. Ammonium nitrogen content was measured using KCl leaching and colorimetric analysis with indophenol blue. Nitrate nitrogen levels were determined using a dual-wavelength UV spectrophotometric method with a correction factor.

### Statistical methods

Experimental data were averaged across three replicates, and statistical analysis was performed using ANOVA with SPSS 27.0. The significance of differences between treatments was evaluated by using the least significant difference (LSD) method. Graphs depicting mean values with standard errors were generated using Origin 2024. Furthermore, the Mantel test was conducted in the R v 4.2.2 (*R Core Team, 2022*) environment using the dplyr, ggcor, and ggplot2 packages. This test determined the significance of various influencing factors on nitrogen mineralization, offering a detailed analysis of their relative importance.

### Calculation of indicators

The formula for calculating the indicator of soil N mineralization characteristics can be written as follows:

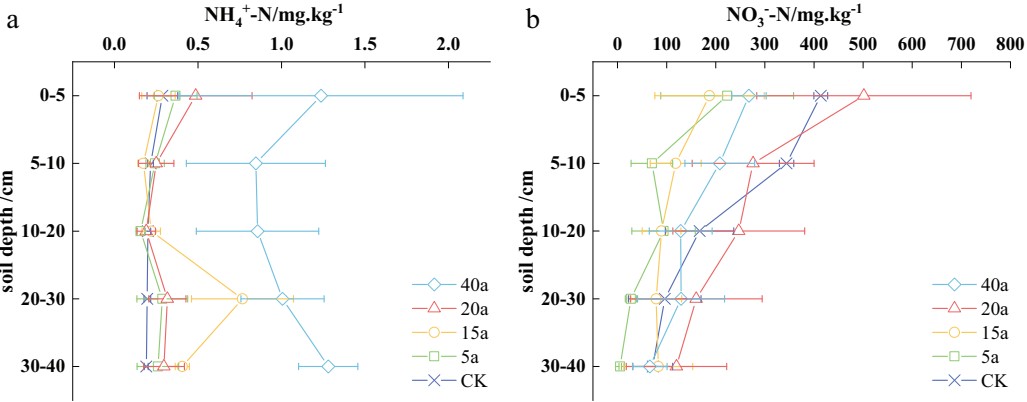

**Figure 2 Effects of vegetation restoration on ammonium nitrogen and nitrate nitrogen.**

$$\text{Net ammonification amount } (mg \cdot kg^{-1}) = NH_4^+ - N \text{ content after culture} \\ - NH_4^+ - N \text{ content before culture}$$

$$\text{Net nitrification } (mg \cdot kg^{-1}) = NO_3^- - N \text{ content after culture} \\ - NO_3^- - N \text{ content before culture}$$

$$\text{Soil mineral nitrogen content } (mg \cdot kg^{-1}) = NH_4^+ - N + NO_3^- - N$$

$$\text{Net mineralization } (mg \cdot kg^{-1}) = \text{soil mineral N content after incubation} \\ - \text{ soil mineral N content before incubation}$$

$$\text{Net nitrogen mineralization rate } (mg/kg \cdot d) = \frac{\text{net mineralisation } (mg \cdot kg^{-1})}{\text{culture days}}$$

$$\text{Net ammonification rate } (mg/kg \cdot d) = \frac{\text{net ammonification amount } (mg \cdot kg^{-1})}{\text{culture days}}$$

$$\text{Net nitrification rate } (mg/kg \cdot d) = \frac{\text{net nitrification amount } (mg \cdot kg^{-1})}{\text{culture days}}$$

## RESULTS AND ANALYSIS

### Effects of vegetation restoration on inorganic nitrogen

The soil $NH_4^+$-N content varied from 0.19 to 1.28 mg·kg$^{-1}$ across different vegetation restoration sites, with the highest concentration observed in the 40-year-old woodland. In particular, the soil's ammonium nitrogen values were determined as 0.84 to 1.28 mg·kg$^{-1}$ for 40-year-old woodlands, 0.19 to 0.48 mg·kg$^{-1}$ for 20-year-old shrubs, 0.17 to 0.76 mg·kg$^{-1}$ for 15-year-old shrub grasslands, 0.15 to 0.36 mg·kg$^{-1}$ for 5-year-old grasslands, and 0.19 to 0.28 mg·kg$^{-1}$ for cultivated areas (Fig. 2A).

In the top 5 cm of soil, the ammonium nitrogen content ranged from 0.26 to 1.23 mg·kg$^{-1}$. Over 40 years, the woodland area demonstrated a notable increase in soil ammonium nitrogen, reaching 1.23 mg·kg$^{-1}$, which was 3.31 times higher than that found in the cultivated land. The sequence of soil ammonium nitrogen levels can be written as follows: 40-year-old woodland > 20-year-old shrub > 5-year-old grassland > cultivated land > 15-year-old shrub grassland. The described pattern remained consistent in the 5–10 cm soil depth. In the 10–20 cm stratum, the order shifted to the following: 40-year-old woodland > 15-year-old shrubland > cultivated land > 20-year-old shrub > 5-year-old grassland. At greater depths, in the 20–30 and 30–40 cm stratums, the sequence changed to the following: 40-year-old woodland > 15-year-old shrubland > 20-year-old shrub > 5-year-old grassland > cultivated land, with increases compared to cultivated land of 4.89, 2.03, 0.58, and 0.41 times, respectively.

The nitrate nitrogen content in the soil ranged from 5.58 to 501.56 mg·kg$^{-1}$ across various vegetation restoration scenarios. In particular, the highest soil nitrate nitrogen content was observed in the 20-year-old shrubland. The levels of soil nitrate nitrogen in a woodland of 40a, shrub of 20a, shrub grassland of 15a, grassland of 5a, and cultivated land varied from 65.9–267.54, 120.02–501.56, 82.88–187.32, 5.58–223.3, and 71.56–413.96 mg·kg$^{-1}$, respectively (Fig. 2B).

In the 0–5 cm soil stratum, the soil nitrate-nitrogen content ranged from 187.32 to 501.56 mg· kg$^{-1}$. The nitrate nitrogen content in the shrub soil after 20 years was significantly higher than that in cultivated land, reaching 501.56 mg·kg$^{-1}$. This value represented a 21.16% increase compared to the nitrate nitrogen content present in the cultivated land. The following is the observed ranking of nitrate nitrogen performance: shrubs after 20 years > cultivated land > woodland after 40 years > grassland after 5 years > shrubland after 15 years. In the 5–10 cm soil stratum, the order of nitrate nitrogen content can be written as cultivated land > shrubs after 20 years > woodland after 40 years > shrubland after 15 years > grassland after 5 years. In the 10–20 cm soil stratum, the nitrate nitrogen performance is similar to that of the 0–5 cm stratum. In the 20–30 cm soil stratum, the following is the nitrate nitrogen performance: shrubs after 20 years > cultivated land > woodland after 40 years > shrub grassland after 15 years > grassland after 5 years. Finally, in the 30–40 cm soil stratum, the ranking of nitrate nitrogen performance was observed as shrubs after 20 years > shrub grassland after 15 years > cultivated land > woodland after 40 years > grassland after 5 years. Overall, shrub growth in the 0–40 cm soil depth demonstrated a 19.38% increase over 20 years compared to cultivated land.

## Effects of vegetation restoration on nitrogen mineralization
### Variation characteristics of soil ammonium nitrogen
The nitrogen content of $NH_4^+$-N in the soil varied from 0.09 to 4.19 mg·kg$^{-1}$ depending on the stage of vegetation restoration. Specifically, the concentrations varied as follows: 0.25 to 4.19 mg·kg$^{-1}$ in a 40-year-old forest, 0.19 to 2.54 mg·kg$^{-1}$ in a 20-year-old shrub area, 0.15 to 2.18 mg·kg$^{-1}$ in a 15-year-old grassland, 0.10 to 1.13 mg·kg$^{-1}$ in a 5-year-old field, and 0.09 to 1.17 mg·kg$^{-1}$ in farmland. The $NH_4^+$-N levels in the soil showed a cyclical pattern, characterized by alternating phases of increase, decrease, increase, decrease, and eventual

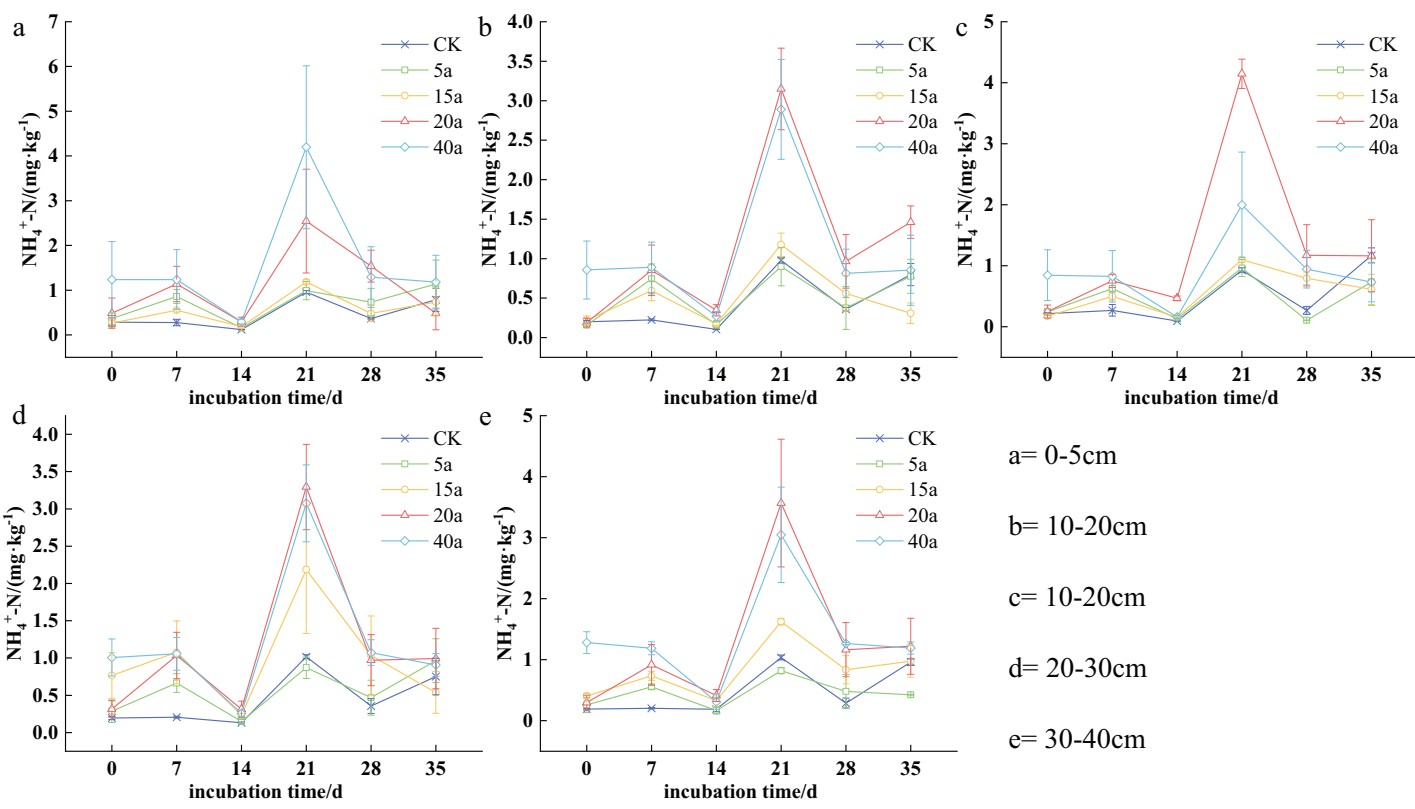

**Figure 3 (A–E) Characteristics of soil ammonium nitrogen changes under vegetation restoration.**

stabilization over time. After 7 days of cultivation, significant differences were observed between various vegetation types and cultivated land. By the 14th day, the overall content decreased but increased again by the 21st day. Furthermore, $NH_4^+$-N concentrations were higher in the 0–10 cm topsoil stratum compared to the deeper stratum (10–20, 20–30, 30–40 cm).

In the 0–5 cm soil layer, $NH_4^+$-N levels reached their lowest point on the 14th day, with no significant statistical differences observed compared to other sampling days. The concentration was found to be highest on the 21st day. The ranking of soil $NH_4^+$-N was as follows: woodland (40a) > shrub (20a) > shrub grassland (15a) > grassland (5a) > cultivated land. Specifically, woodland (40a) and shrub (20a) increased by 3.39 times and 1.66 times, respectively, compared to cultivated land. In the 5–10 cm soil layer, after the 21st day, the order of $NH_4^+$-N concentrations was as follows: shrub (20a) > woodland (40a) > shrub grassland (15a) > cultivated land > grassland (5a). In particular, shrub (20a) and woodland (40a) increased by 3.47 times and 1.15 times compared to cultivated land. After 21 days in the 10–20 cm soil stratum, the order of soil $NH_4^+$-N was determined as shrub (20a) > woodland (40a) > shrub grassland (15a) > grassland (5a) > cultivated land. In this case, shrub (20a) and woodland (40a) demonstrated 2.22 times and 1.95 times increase compared to the cultivated land. This trend continued in the 20–30 cm soil layer, where $NH_4^+$-N levels in the 20-year-old shrub and 40-year-old woodland were 2.23 times and 2.02 times higher, respectively, compared to the cultivated land. Finally, in the 30–40 cm

soil stratum, the trend remained consistent with previous stratums, where shrub (20a) and woodland (40a) increased by 2.44 and 1.94 times upon comparison with cultivated land (Fig. 3).

### Variation characteristics of soil nitrate nitrogen

Soil nitrogen content varied significantly among different vegetation restoration types, ranging from 14.20 to 868.06 mg·kg$^{-1}$. Specifically, the levels ranged from 53.84 to 868.06 mg·kg$^{-1}$ in 40a woodland, 120.02 to 680.38 mg·kg$^{-1}$ in 20a shrubs, 72.94 to 454.58 mg·kg$^{-1}$ in 15a shrub grassland, 14.08 to 676.29 mg·kg$^{-1}$ in 5a grassland, and 34.26 to 560.42 mg·kg$^{-1}$ in cultivated land. As time progressed, soil nitrogen content generally increased with the duration of cultivation, peaking on the 14$^{th}$ day. Furthermore, nitrogen levels were consistently higher in the top 0–10 cm soil stratum compared to the deeper strata (10–20, 20–30, and 30–40 cm).

The highest concentration of soil $NO_3^-$-N was observed on the 14$^{th}$ day of cultivation. In the top 0–5 cm soil stratum, soil $NO_3^-$-N levels were observed as follows: 40a woodland > 20a shrubs > cultivated land > 15a shrub grassland > 5a grassland, with 40a woodland and 20a shrubs demonstrating a respective increase of 0.44 and 0.39 times compared to cultivated land. In the case of the 5–10 cm soil stratum, the order shifted to the following: 20a shrubs > cultivated land > 40a woodland > 15a shrub grassland > 5a grassland, with shrubs showing a 0.13 times increase compared to cultivated land in 20a. The described trend persisted in the 10–20 cm soil stratum, with shrubs showing a 0.46 times increase compared to cultivated land in 20 years. In the 20–30 cm soil stratum, soil $NO_3^-$-N levels ranked as follows: 20a shrubs > 40a woodland > 15a shrubland > cultivated land > 5a grassland, where 40a woodland and 20a shrubs demonstrated a respective increase of 0.71 and 0.92 fold compared to cultivated land. The pattern remained consistent in the 30–40 cm soil stratum, with 40a woodland and 20a shrubs showing a 0.17 and 1.36 times increase, respectively, compared to the cultivated land (Fig. 4).

### Vegetation restoration on net ammonification rate

The results depicted in Fig. 5 revealed that the soil's net ammoniation rate followed a fluctuating pattern over time: it initially decreased, then increased, subsequently declined, and finally stabilized. On the 14$^{th}$ day, the soil's net ammoniation rate reached its lowest level during cultivation, while the highest concentration was observed on the 21$^{st}$ day. Notably, the ammoniation levels on the 35$^{th}$ day were found to be lower than those observed on the 7$^{th}$ day. With the increase in the cultivation period, the soil's ammoniation impact was reduced, resulting in a decrease in the net ammoniation rate. Throughout cultivation, the net soil mineralization rate increased in the following order: 40a woodland, cultivated land, 5a grassland, 15a shrub grassland, and 20a shrub grassland, with an average value of 7.36, 11.18, 21.11, 21.71, and 58.26 mg/(kg·d), respectively. Compared to cultivated land, the net mineralization rate increased by 0.94 times for 15-year-old shrub grassland and 4.21 times for 20-year-old shrub grassland.

During the first 7 days of planting, the soil net nitrification rate (AR) size at different depths followed a consistent order: 20a shrubs > 5a grassland > 15 shrubland > cultivated

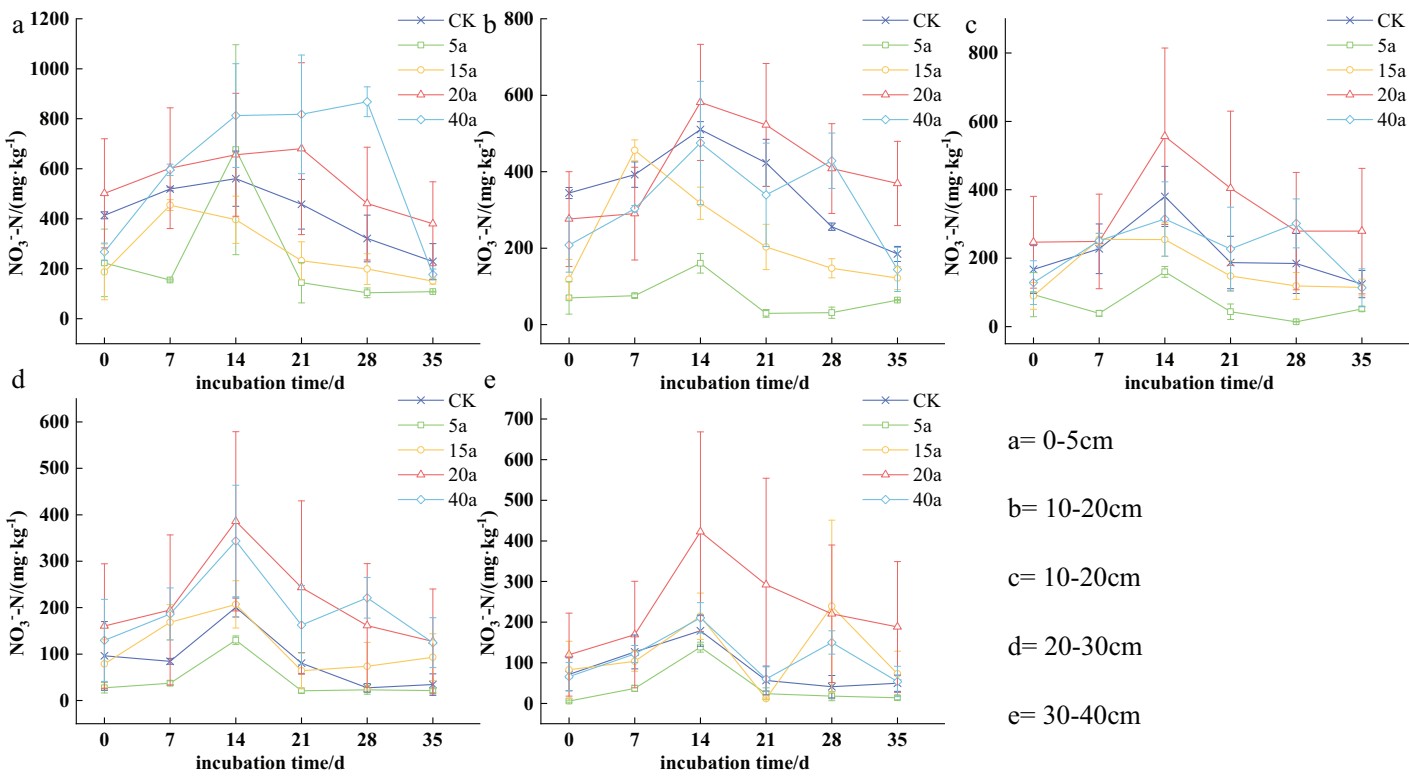

**Figure 4** (A–E) Characteristics of soil nitrate-nitrogen changes under vegetation restoration.

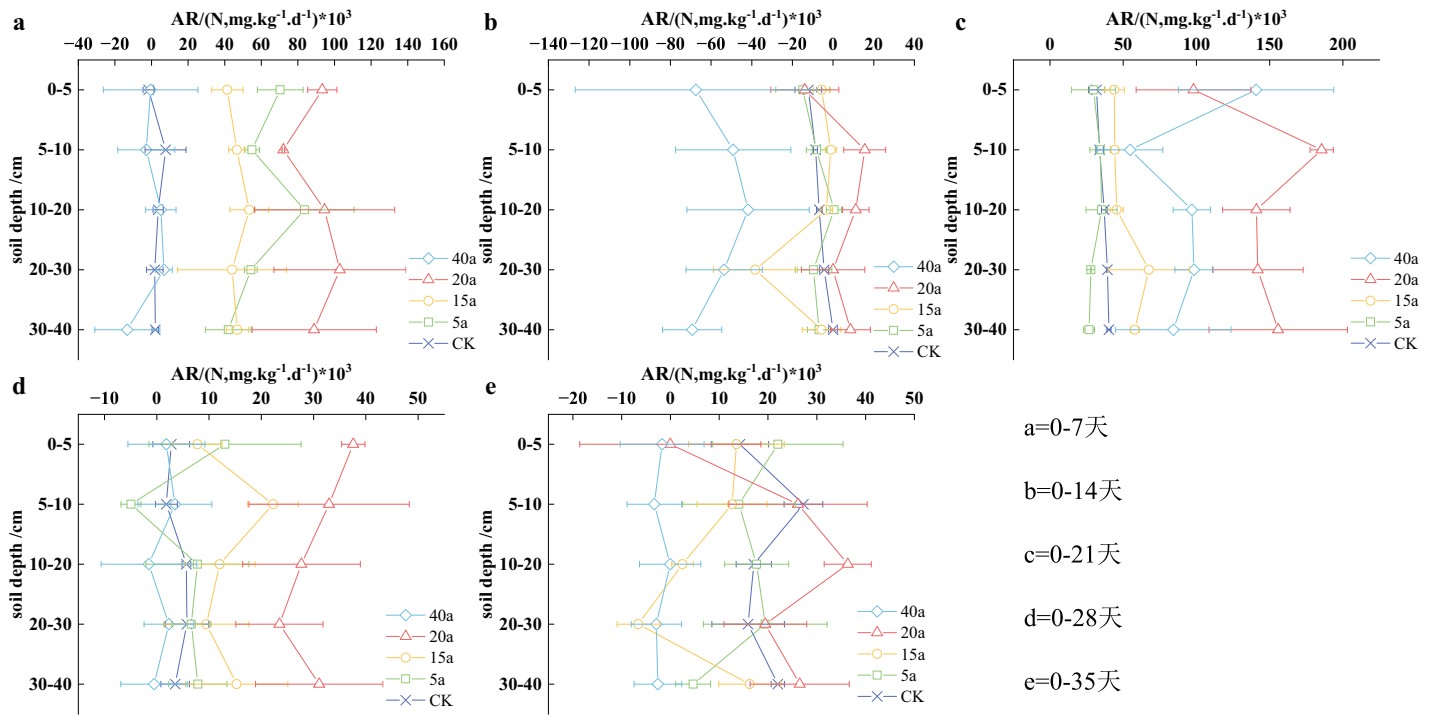

**Figure 5** (A–E) Characteristics of net soil ammonification rate under vegetation restoration.

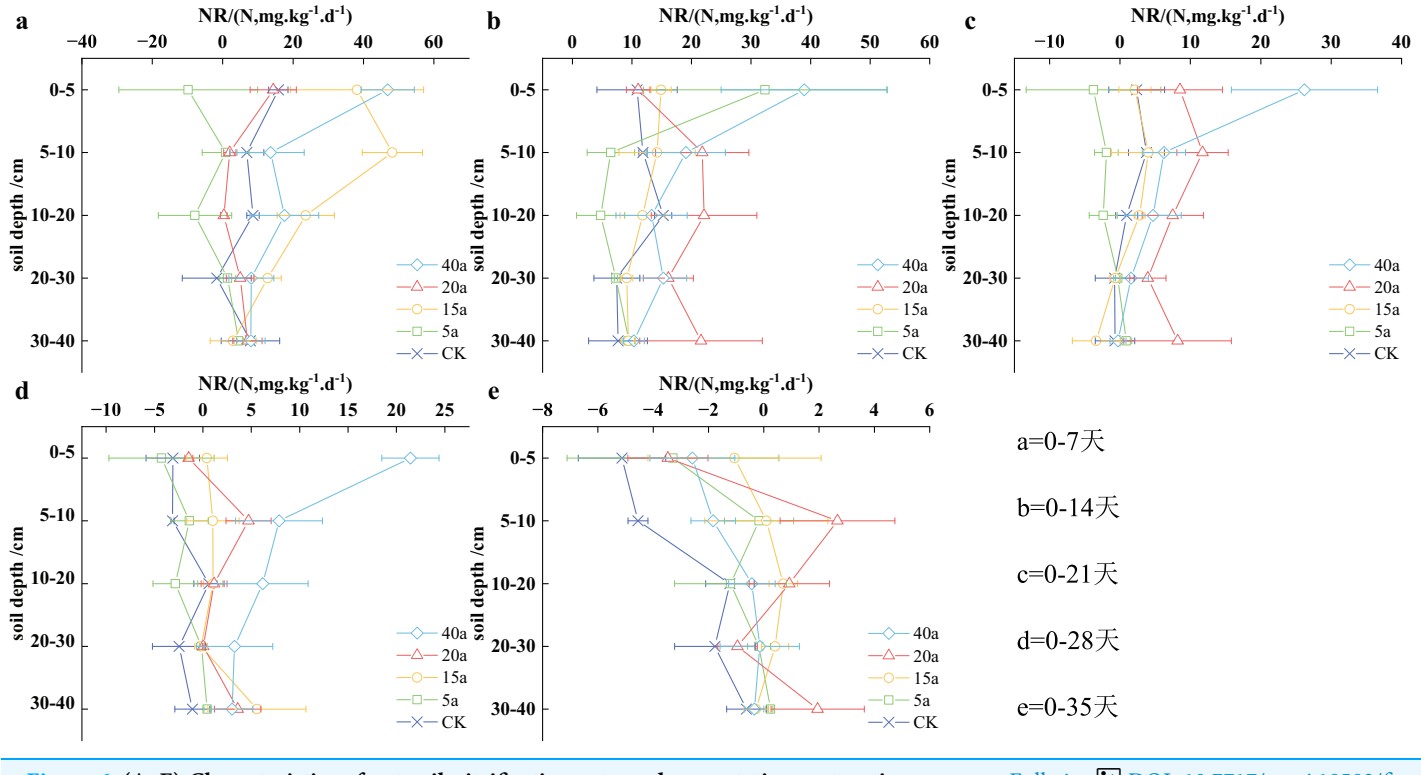

**Figure 6 (A–E) Characteristics of net soil nitrification rate under vegetation restoration.**

land > 40a woodland. Specifically, 20a shrubs, 5a grassland, and 15a shrub vegetation demonstrated significantly higher measurements than cultivated land. With the increase in cultivation to 14 days, woodland in 40a showed the lowest measurement, compared to other vegetation types. On the other hand, 20a shrubs displayed the highest measurement, with no significant differences observed among the other vegetation types. After 21 days of cultivation, the soil AR size ranked as follows: 20a shrub > 40a woodland > 15a shrubland > cultivated land > 5a grassland, with a notable difference between shrubs and cultivated land in 20a. By the 28th day, the sequence of soil AR size remained consistent across the strata: 20a shrubs > 15a shrubland > 5a grassland > cultivated land > 40a woodland, with considerable differences observed between shrubs and cultivated land in 20a. With increase in planting till day 35, the soil AR size in the 0–5 cm stratum was determined as follows: 5a grassland > 15a shrub vegetation > cultivated land > 20a shrubs > 40a woodland, whereas in the 5–10, 10–20, 20–30, and 30–40 cm stratums, the arrangement was observed as follows: 20a shrubs > cultivated land > 5a grassland > 15 shrub vegetation > 40a woodland.

### Vegetation restoration on net nitrification rate

Figure 6 illustrates a progressive decrease in the soil net nitrification rate over time. The peak rate was observed at 7 days of cultivation, with the lowest rate recorded at 35 days. Furthermore, the soil net nitrification rate consistently decreased over the culture period, indicating a reduction in soil nitrification with prolonged cultivation time. Throughout the cultivation process, the net soil mineralization rate decreased sequentially in a 40-year-old

woodland, 15-year-old shrubland, a 20-year-old shrub, cultivated land, and a 5-year-old grassland, with an average value of 10.63, 7.91, 6.81, 2.94, and 1.16 mg/(kg·d) respectively. Woodland aged 40 years, shrubland aged 15 years, and shrubs aged 20 years demonstrated respective increases of 2.62 times, 1.69 times, and 1.32 times, compared to cultivated land.

During the initial 0–7 days of soil incubation, the size of soil NR in the 0–5 cm stratum followed the order of 40a woodland > 15a scrub grassland > cropland > 20a shrub > 5a grassland, with 40a woodland showing significantly higher N nitrification rate (NR) than cropland. In the subsequent 5–10, 10–20, and 20–30 cm stratum, the soil NR size was ranked as follows: 15a scrub grassland > 40a woodland > cropland > 20a shrub > 5a grassland, with 15a scrub grassland, demonstrated remarkably higher NR than cropland. However, in the 30–40 cm stratum, no significant differences were observed in vegetation recovery. For the case of 0–14 days of incubation, in the 0–5 cm stratum, the soil NR size was determined as follows: 40a woodland > 5a grassland > 15a shrub meadow > 20a shrub > cropland, with 40a woodland showing significantly higher NR than cropland. In the subsequent stratum (5–10, 10–20, and 20–30 cm), the soil NR size showed the following sequence: 20a shrub > 40a woodland > cropland > 15a shrub meadow > 5a grassland, with 20a shrub showing significant superiority over cropland. In the 30–40 cm stratum, no significant differences were observed among the various types of vegetation restoration.

During the initial 0–21 days of incubation, the soil NR size in the 0–5 cm stratum was ranked as follows: 40a woodland > 20a shrub > 15a shrub meadow > cropland > 5a grassland, with 40a woodland demonstrating significant superiority over cropland. In the subsequent stratum (5–10, 10–20, 20–30, and 30–40 cm), the overall trend for soil NR size can be written as 20a shrub > 40a woodland > 15a shrub meadow > cropland > 5a grassland. Lastly, at 0–28 days of incubation, the 0–5 cm stratum showed a soil NR size sequence of 40a woodland > 15a shrub meadow > 20a shrub > cropland > 5a grassland, with 40a woodland showing significantly higher NR than cropland. In the subsequent stratum (5–10, 10–20, and 20–30 cm), the overall soil NR size was ranked as follows: 40a woodland > 20a shrub > 15a shrub meadow > cropland > 5a grassland. In the stratum of soil measuring 30–40 cm, there were no significant differences observed in plant life. Following an incubation period of 0–35 days, the ranking of NR size in the soil stratum of 0–5 cm was observed in the following order: 15a meadow of shrubs > 40a forested area > 20a shrubbery > 5a grassy meadow > cultivated land. For the stratum of soil measuring 5–10 and 10–20 cm, the NR size ranking can be written as follows: 20a shrubbery > 15a meadow of shrubs > 5a grassy meadow > 40a forested area > cultivated land. However, in the stratum of soil measuring 20–30 and 30–40 cm, no significant disparities were observed in vegetation regeneration.

### Vegetation restoration on net nitrogen mineralization rate

The nitrogen mineralization rate in the soil reflects the changes in inorganic nitrogen over time. This research evaluated the nitrogen mineralization rate at varying time intervals, ranging from 0–7 to 0–35 days. Figure 7 illustrates a distinct decrease in nitrogen mineralization throughout vegetation restoration, with a peak value of 46.86 ± 7.55 mg/(kg·d) observed on the 7[th] day in a 40-year-old forest area. The ranking of nitrogen

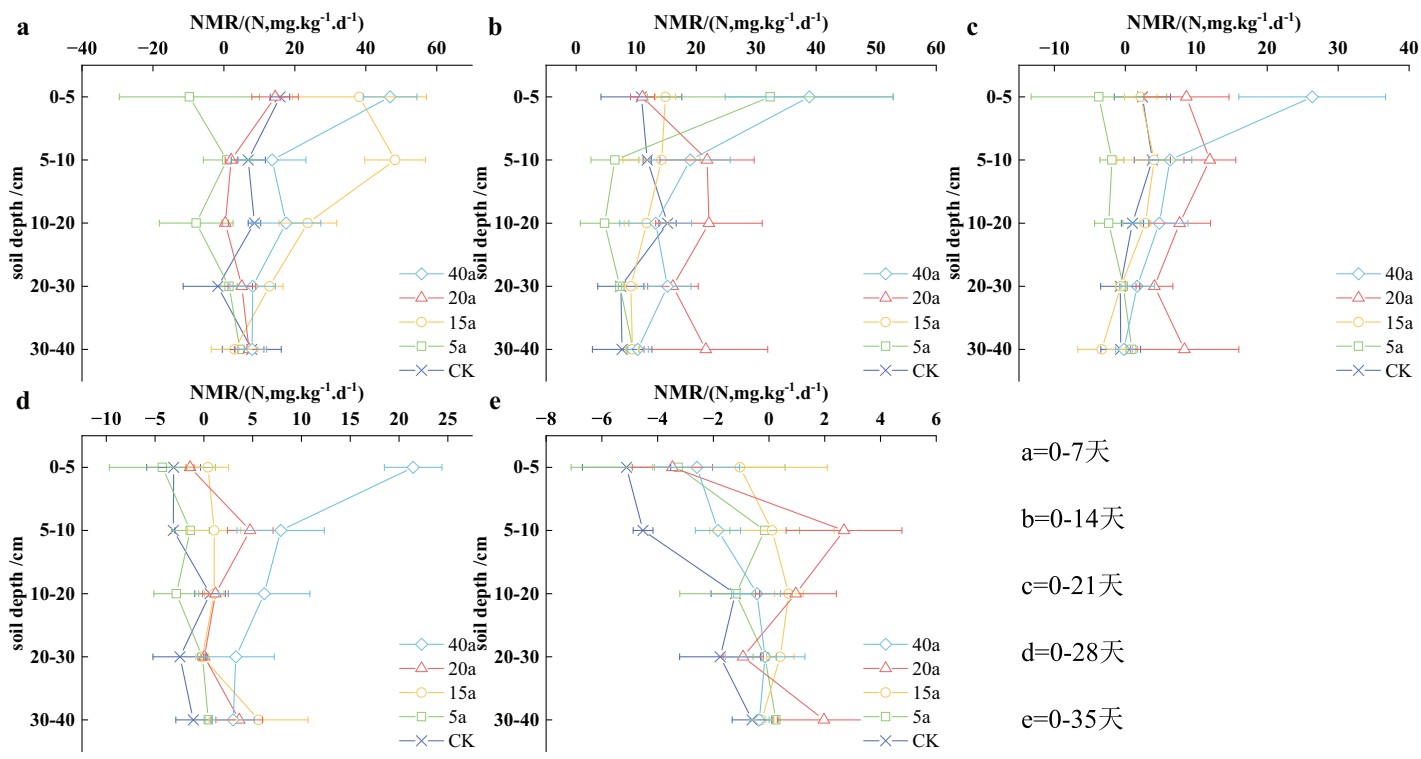

a=0-7天

b=0-14天

c=0-21天

d=0-28天

e=0-35天

**Figure 7 (A–E) Characteristics of net soil nitrogen mineralisation rate under vegetation restoration.**

mineralization rates in the 0–40 cm soil stratum can be written as follows: shrub grassland aged 15 years > woodland aged 40 years > cultivated land > shrubland aged 20 years > grassland aged 5 years. The reduction in nitrogen mineralization was associated with longer cultivation periods, resulting in a slower nitrogen mineralization process in the soil.

Throughout the cultivation period, the nitrogen mineralization rates in woodland aged 40 years, shrubland aged 15 years, shrubland aged 20 years, grassland aged 5 years, and cultivated land decreased progressively, with average values of 10.64, 7.93, 6.87, 2.95, and 1.18 mg/(kg·d) respectively. In comparison, nitrogen mineralization rates in 40-year-old woodland, 15-year-old shrubland, and 20-year-old shrubland were determined to be 8.01, 5.72, and 4.82 times higher, respectively than those observed in cultivated land. Overall, vegetation restoration has the potential to significantly enhance soil nitrogen mineralization rates.

During the initial 7 days of cultivation, the highest soil nitrogen mineralization rate was recorded at 46.86 ± 7.55 mg/(kg·d) in a 40-year-old woodland area, while the lowest value was found to be negative in a 5-year-old grassland region. In the 0–40 cm soil stratum, the NMR values showed the following order: 15a shrub grassland > 40a woodland > cultivated land > 20a shrub > 5a grassland. Specifically, in the 0–5 cm soil stratum, the sequence of soil NMR values can be written as 40a woodland > 15a shrubland > cultivated land > 20a shrub > 5a grassland, with 40a woodland showing significantly higher values compared to cultivated land and 5a grassland demonstrating negative values. In the 5–10

and 10–20 cm soil stratum, the soil NMR trends were determined to be consistent with those observed in the 0–40 cm soil stratum. However, no significant difference was observed between the 20–30 and 30–40 cm soil stratum.

During the initial 2 weeks of planting, the soil NMR levels varied from 38.85 ± 14.00 mg/(kg·d) in the 40a forest area to 4.74 ± 4.04 mg/(kg·d) in 5a grassy terrain. The sequence of nitrogen mineral rates in the 0–40 cm soil stratum was determined as follows: 40a woodland > 20a shrub > 5a grassland > 15a shrubland > farmland. In the top 5 cm soil stratum, the soil NMR hierarchy was determined as follows: 40a woodland > 5a grassland > 15a shrubland > 20a shrubs > farmland. For the case of 5–10 cm soil stratum, the soil NMR performance showed the following ranking: 20a shrubs > 40a woodland > 15a shrubland > farmland > 5a grassland. In the 10–20 cm soil stratum, the soil NMR pattern demonstrated the following order: 20a shrubs > farmland > 40a woodland > 15a shrubland > 5a grassland.

Significant differences were observed between 20a shrubland, 40a woodland, and farmland in the 20–30 and 30–40 cm soil segments.

Within the first 21 days of planting, the highest soil NMR value was recorded as 26.33 ± 10.34 mg/(kg·d) in the 40a woodland. In the 0–40 cm soil stratum, the following sequence of soil NMR values was observed: 40a woodland > 20a shrubs > land under cultivation > 15a shrub grassland > 5a grassland. For the soil stratum of 0–5 cm, the soil NMR values were found to be similar to those observed in the 0–40 cm stratum, with 40a woodland demonstrating significantly higher values compared to cultivated land. In the soil stratum of 5–10, 10–20, 20–30, and 30–40 cm, the following sequence of soil NMR values was observed: 20a shrub > 40a woodland > 15a shrub grassland > land under cultivation > 5a grassland.

Throughout the first 28 days of plant growth, the 40-acre woodland area demonstrated significantly larger soil NMR values in the top 5 cm stratum compared to other vegetation types. As the cultivation increased till day 35, the soil NMR dimensions in the top 5 cm stratum were ranked in the following order: shrub grassland spanning 15 acres > woodland area of 40 acres > shrub area of 20 acres > grassland area of 5 acres > cultivated land. In the 5–10, 10–20, 20–30, and 30–40 cm soil stratum, the overall soil NMR dimensions were observed in the following order: shrub area of 20 acres > shrub grassland spanning 15 acres > woodland area of 40 acres > grassland area of 5 acres > cultivated land.

During the incubation period across different plant regenerations, soil ammonification rates were significantly lower than nitrification rates, demonstrating an increasing trend that contrasted with the declining pattern of soil nitrification. The predominant net mineralization form in the soil was nitrification, consistent with established findings on soil nitrification processes.

### Effects of environmental factors on vegetation restoration

Analysis of the Mantel test revealed the positive correlation between AR and total nitrogen (TN), total phosphorous (TP),  total potassium (TK), ammonium nitrogen (AN), AK, soil organic carbon (SOC), silicon (Si), $NH_4^+$-N; NR and TN, TP, TK, Si, BD, STP, SMC; NMR and TN, TP, TK, AN, AK, SOC, Si, Sa, pH. In contrast, AR, NR, and NMR showed negative

correlations with $NO_3^-$-N and Ap. The findings suggest that soil mineralization processes in the study area were primarily influenced by TN, TP, TK, and Si, with ammonium ($NH_4^+$-N) having the most significant effect on AR, BD affected the NR, and ammonium nitrogen (AN) and SOC influenced the NMR. Moreover, $NH_4^+$-N showed a positive correlation with TK, AK, and C, while $NO_3^-$-N was positively associated with TN, AP, and SOC.

## DISCUSSION

### Effects of vegetation restoration on inorganic nitrogen

Subtropical and tropical forests are generally considered phosphorus-limited, while temperate and boreal forests are often nitrogen-limited (*Elser et al., 2007*). Recent research by *Zhang et al. (2015)* and *Lan, Hu & Fu (2020)* has shown that karst landscapes in subtropical regions can exhibit nitrogen-limited conditions as vegetation begins to regenerate. The detection of $NO_3^-$-N as the primary form of inorganic nitrogen in this study was consistent with the results of *Hu et al. (2021)*, showing a 19.38% increase in shrub areas compared to cultivated areas. The observed increase in $NO_3^-$-N was associated with vegetation recovery, which enhanced the accumulation of plant debris and roots in the soil. This process improves soil permeability and stimulates bacterial and microbial activities, ultimately leading to higher nitrogen concentrations. The higher levels of $NO_3^-$-N in woodlands compared to grassland samples observed in this study supported the findings reported by *Xing et al. (2013)*, contrasting grassland ecosystems with a wide variety of trees, shrubs, and grasses (*Li et al., 2019*; *Dong et al., 2022a*). Woodlands often contain more apoplastic substances with lesser C/N ratios, leading to the presence of greater mineral N in the surface soil (*Pang et al., 2020*; *Babur et al., 2022*). The research findings revealed a significant variation in nitrate N levels between the upper and lower soil strata. (*Karki, Bargali & Bargali, 2021*; *Dong et al., 2022b*; *Siwach et al., 2023*). This variation was attributed to the rich oxygen environment, abundant organic matter, and diverse array of microorganisms in the topsoil, which facilitated the nitrification process. In the soil stratum ranging from 0–40 cm, the ranking of soil ammonium nitrogen levels was as determined as follows: woodland > shrub-grassland > shrub > grassland > arable land, with increases of 4.89, 2.03, 0.58, and 0.41 times respectively compared to arable land. These variations were linked to the continuous decomposition of plant matter during the process of vegetation restoration, ongoing mineralization of organic nitrogen in the soil, accumulation of inorganic nitrogen, and the increase in $NH_4^+$-N content.

### Effects of vegetation restoration on nitrogen mineralization

Restoring vegetation significantly enhances the uptake and consumption of ammonium and nitrate nitrogen in the soil (*Maslov & Maslova, 2022*; *Wang et al., 2023*). According to *Loeb et al. (2009)*, the rate of nitrogen mineralization is crucial for supplying these essential nutrients. Over time, soil accumulates various nitrogen species, leading to substantial nitrogen reserves. The rate of net mineralization is a key metric for the assessment of soil nitrogen effectiveness (*Zhang et al., 2021b*). They also revealed that inorganic nitrogen concentrations in the soil varied from 14.50 to 869.36 mg·kg$^{-1}$ across different vegetation

restoration conditions. As the incubation period increased, the levels of inorganic nitrogen in the soil generally showed a pattern of increase, followed by a decrease and eventual stabilization, peaking on the 14th day, consistent with the trends observed in nitrate nitrogen. The adsorption and utilization of $NO_3^-$-N in the soil are restricted by its negatively charged nature, resulting in the majority of $NO_3^-$-N remaining in the soil solution without being effectively consumed. Soil nitrification processes consume ammonium nitrogen from the soil and external fertilizers, reducing the loss due to ammonia volatilization and leading to $NO_3^-$-N accumulation in the soil. The study highlighted the presence of higher concentrations of inorganic nitrogen in the upper 0-10 cm soil stratum compared to the deeper stratum. Surface soils have a higher capacity to absorb external organic N, leading to more rapid accumulation. In contrast, with increase in soil depth, permeability generally decreases, which slows down the aging and decomposition processes of soil organic matter (*Qiu et al., 2021*). This results in decreased organic matter availability for decomposition and plant uptake, causing a reduction in microbial populations and activity, which could potentially lower the rate of N mineralization (*He et al., 2021*). Soil surface temperature variations significantly affect microbial activity, whereas deeper soil strata are less influenced by these changes (*Naylor, McClure & Jansson, 2022*). The research demonstrates that $NO_3^-$-N is the primary form of inorganic nitrogen. Initially, vegetation absorbs $NO_3^-$-N, resulting in a significantly lower net residual of $NO_3^-$-N in areas with dense vegetation compared to those with sparse vegetation (*Li et al., 2017*). The loss rate of $NO_3^-$-N exceeded that of $NH_4^+$-N, while the relatively high levels of $NH_4^+$-N in soils across varying locations contributed to the sustained nitrogen content in the soil (*Sainju et al., 2006*). Soil inorganic N levels increased gradually during vegetation mineralization recovery but decreased with increasing the recovery period. This trend resulted from the gradual stabilization of organic matter and apoplastic material in the soil, coupled with decreased microbial activity during the process of vegetation restoration. Therefore, the mineralization processes slowed down, leading to varying concentrations of inorganic nitrogen over time as the restoration period progressed.

## Factors affecting soil nitrogen mineralization

Vegetation plays an essential role in the nitrogen cycle and is vital for soil health (*Manral et al., 2020*). It enhances water retention, improves soil aeration and infiltration rates, and contributes to better soil texture (*Zhang et al., 2021a*). These improvements can directly influence the structure and operation of plants (*Pandey et al., 2023*). Furthermore, the contributions from vegetation debris and decomposition processes play a crucial role in determining soil nutrient levels (*Awasthi et al., 2022*; *Pandey et al., 2024*). Land use patterns considerably affect the fertility and stability of an ecosystem, serving as a crucial source of nutrients due to their rapid turnover (*Padalia et al., 2018*). Moreover, plants, along with cultural practices, can alter the soil environment by influencing the microclimate and generating detritus (*Trentini et al., 2018*). They contribute to nutrient redistribution (*Wu et al., 2021*), enhance $N_2$ fixation (*Li et al., 2021*), improve soil biota (*Cai et al., 2022*), and influence soil physicochemical characteristics (*Qiu et al., 2022*; *Lyu*

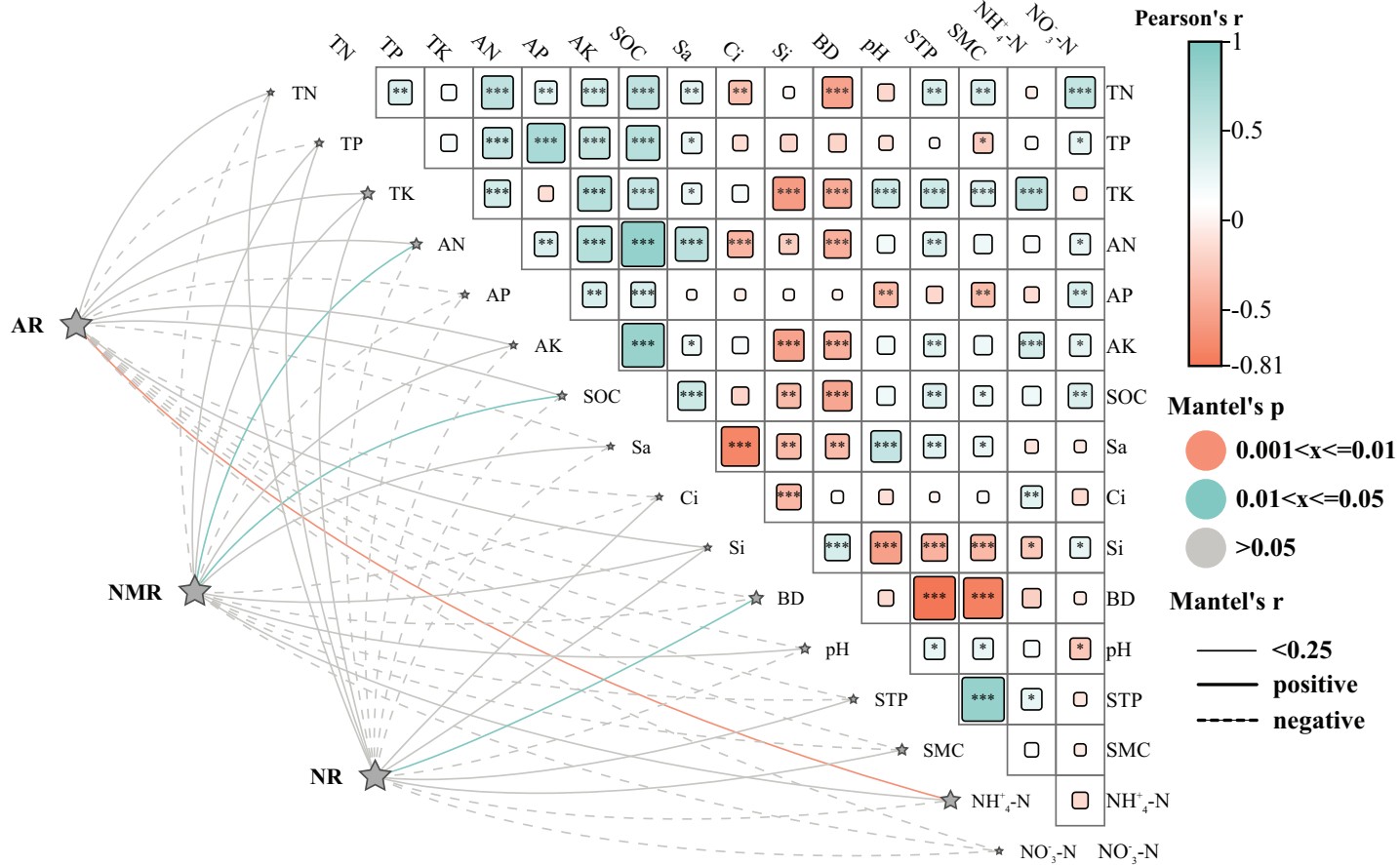

**Figure 8 Effects of environmental factors on nitrogen mineralisation.** Abbreviations: The heat map's rectangular shapes illustrate the correlations between soil physicochemical factors. The line's thickness represents Mantel's r-test correlation coefficient magnitude, with solid lines indicating positive correlations and dashed lines representing negative correlations. The line colors in the heat map represent Mantel's p-test value (grey signifies *$P < 0.05$, cyan denotes **$P < 0.01$, orange represents ***$P < 0.001$). These correlations examine various organic nitrogen fractions and environmental factors such as AR (net ammonification rate), NR (net nitrification rate), NMR (net nitrogen mineralization rate), SOC (soil organic carbon), pH, BD (looseness), STP (total porosity), Cl (clayey grains), Si (silt grains), Sa (sandy grains), SMC (water content), TN (total nitrogen), AN (alkali dissolved nitrogen), AP (quick phosphorus), AK (quick potassium), $NH_4^+$-N (ammoniacal nitrogen), and $NO_3^-$-N (nitrate-nitrogen).

*et al., 2023*). The Mantel test analysis revealed positive correlations of TN, TP, TK, and Si with soil AR, NR, and NMR (Fig. 8), consistent with the findings reported by *Li et al. (2019)*. However, pH showed a negative correlation with AR and NR, indicating that substrate abundance and favorable conditions can enhance soil N mineralization (*Li et al., 2018*), a critical factor in regulating soil N availability (*Wei et al., 2011*). N transformations including mineralization and nitrification greatly influence soil N availability. Sa showed a negative correlation with AR and NR, aligning with studies that suggest clay-rich, fine-textured soils typically contain a higher amount of microbial biomass, organic carbon, and nitrogen compared to coarse-textured soils, thereby enhancing overall N mineralization (*Ding et al., 2021*). Total N mineralization was significantly higher in soils with elevated levels of fines and clays (*Elrys et al., 2023*). SMC was negatively correlated with soil $NO_3^-$-N, possibly due to increased $NO_3^-$-N depletion in soils with higher moisture

contents (*Srivastava et al., 2015*). Furthermore, alkaline soils hinder the decomposition of organic matter, leading to reduced soil N mineralization (*Beltran-Hernandez et al., 1999*). This was consistent with the observed negative relationship between pH and soil AR and NR parameters. Soil inorganic N levels play a crucial role in soil nutrient dynamics, with soil NMR indirectly affecting SOC and serving as an important indicator of soil fertility in revegetated ecosystems (*Wei et al., 2009*). Factors affecting soil N transformations vary across ecosystems, primarily due to variations in climate, vegetation, and land use history (*Burke, 1989*; *Li et al., 2014*; *Maithani et al., 1998*).

## CONCLUSION

(1) Over 20 years, shrubs demonstrated a 19.38% increase in inorganic nitrogen at the 0–40 cm depth, primarily in the form of nitrate nitrogen, compared to cultivated areas.

(2) As the duration of incubation increased, soil $NH_4^+$-N levels showed a distinct pattern: an initial rise, followed by a decrease, a subsequent spike, another decrease, and finally, stabilization. On the 14th day of incubation, the $NH_4^+$-N content reached its lowest point, while it peaked on the 21st day. In contrast, the soil $NO_3^-$-N and total inorganic nitrogen showed a pattern of increase, decrease, and stabilization, with their highest levels observed on 14 days of incubation. Furthermore, the $NO_3^-$-N, $NO_3^-$-N, and organic nitrogen levels in the upper 0–10 cm stratum of soil exceeded those in the deeper stratum (10–20, 20–30, and 30–40 cm).

(3) Under vegetation restoration, the soil NR and NMR gradually decrease with incubation time. However, soil NR and NMR increased in 15-year shrub, 20-year shrub, and 40-year woodland areas. Moreover, the 15-year shrub and 20-year shrub conditions led to higher soil AR.

(4) The analysis of the Mantel test showed positive correlations between TN, TP, TK, and Si with soil AR, NR, and NMR. Among these, $NH_4^+$-N had the most significant impact on AR, BD had the most pronounced influence on NR, and AN and SOC were identified as the key driving forces of NMR.

### Funding

This work was funded by the Ministry of Agriculture and Rural China (Z2023365) and the National Natural Science Foundation of China (31460133). The funders had no role in study design, data collection and analysis, decision to publish, or preparation of the manuscript.

### Grant Disclosures

The following grant information was disclosed by the authors:
Ministry of Agriculture and Rural China: Z2023365.
National Natural Science Foundation of China: 31460133.

## Competing Interests

The authors declare that they have no competing interests.

## Author Contributions

- Jianghong Wu conceived and designed the experiments, performed the experiments, analyzed the data, prepared figures and/or tables, and approved the final draft.
- Xianghuan Gong conceived and designed the experiments, analyzed the data, authored or reviewed drafts of the article, and approved the final draft.
- Yingge Shu conceived and designed the experiments, authored or reviewed drafts of the article, and approved the final draft.

## Data Availability

The raw data is available in the Supplemental File.

## Supplemental Information

Supplemental information for this article can be found online at http://dx.doi.org/10.7717/peerj.18582#supplemental-information.

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
