# Peer review of "Effects of vegetation restoration in karst areas on soil nitrogen mineralisation"

_PeerJ, doi:10.7717/peerj.18582_

## Round 0.1 · original submission · Major Revisions

Dear Dr. Shu,

I have now received three reviews of your paper. As you will note from them, the reviewers requested and suggested various changes to your paper, related to context, results, statistics and interpretation.

Please revise your paper accordingly, and provide a detailed rebuttal letter when resubmitting your paper.

Sincerely,
Shaw Badenhorst

·

Basic reporting

no comment

Experimental design

Soil sampling up to 40 cm cannot be recognized as a profile study.
Line 145-148, these are your main parameters and how did you measured should be briefly described.
For the exact determination of net mineralization potential of any soil or manures, Stanford and Smith 1972, a classic method is always recommended.

Validity of the findings

see additional comments

Additional comments

This study was conducted to evaluate the impact of restored vegetation on nitrogen mineralization in Karst ecosystem. But this study lack a major problem, why it was conducted? What is the significance and implication of the study? However, besides English must be checked by a fluent speaker, the manuscript needs a thorough revision on the logical structure. In the current version, I am easy to be lost during the reading process. I have some concerns with this manuscript, which need to be justified. Moreover, it seems that authors did not read the manuscript carefully. I suggest revising and rewriting the whole manuscript. The current version of the manuscript is not suitable for publication in “PeerJ” journal. However, few of my concern and comments need to follow.
Results repeatedly discussed in discussion section. In addition, it is strange to see others studies were presented and explained their results, rather, author should present their main findings, explain the mechanisms, and give fewer citations to support your arguments.
Line 21, firstly use full form of nutrient element then used abbreviated i.e. nitrogen (N)
Line 21-22, 35 days are not enough to features out complete N mineralization, because there are some labile, very labile and recalcitrant portion.
Line 25, displayed, use an alternative word,
Line 27-28, subscript and superscript, what is meant by 14th and 21st level? This is unclear. Follow this comment for whole manuscript.
Line 28, once you have used nitrogen (N) , just abbreviated as N, do not write full form, follow this comment for whole manuscript. Please follow the same comment in introduction section and rest of the manuscript
Line 28, Soil NO3--N and inorganic nitrogen levels, what is meant by this? Nitrate-N is not inorganic N?
Result in abstract section, poorly presented, without any conclusive statement and implication of the study.
There is no space in ending sentence and citations in introduction section.
Line 57, I do no think so all these factors have been studied in (Templer et al., 2005). Recheck otherwise add more relevant citations.
Soil sampling up to 40 cm cannot be recognized as a profile study.
Line 145-148, these are your main parameters and how did you measured should be briefly described.
Line 151, what is PE, should be explained……
Line152-153, you did not mention how moisture content was maintained, and temperature was adjusted?
Line 153, pre-cultivation stage, what is this?
For the exact determination of net mineralization potential of any soil or manures, Stanford and Smith 1972, a classic method is always recommended.
Line 170, what is meant by culture?
Line 409, akin???
Line 25 and Line 411, in abstract these are not your results then why you mentioned in the abstract?

·

Basic reporting

General comments
The manuscripts addresses an important ecological issue in an important ecosystem (i.e. restoration of vegetation in the Karst ecosystem and its effect on soil nitrogen mineralization). A sufficiently wide range of relevant soil properties has been measured. However, as explained below, there are serious issues with the selection of soil sampling points, the level of replication and statistical analysis of the data. There are fundamental flaws in the presentation and interpretation of results. Therefore, a meaningful assessment cannot be made of the Discussion and Conclusions.

Specific comments

The manuscript displays correct usage of the English language throughout.

The background and context is adequately presented with sufficient references to relevant literature.

However, Materials and Methods, Results and Conclusions have serious shortcomings. Specific details are given below under Experimental Design and Validity of the Findings.

The figures do not meet the quality standards of PeerJ. Axis labelling is not clear. Figure captions are not sufficiently descriptive. The shared raw data do not provide information about the classification variables (i.e. do not identify the ecosystems and depths) for any independent statistical analysis.

Abstract
Background, rationale and methods are expressed clearly and concisely. A substantial amount of results are given in the abstract. It is preferable if the abstract presents a selected sub-set of key results while leaving the rest in the manuscript text. The abstract should include a clear overarching conclusion section with a brief statement on the broader implications of the key findings of the work.

Introduction
The Introduction is well-written with a clear reasoning of the rationale of the work. It is supported with an adequate review of recent and relevant literature. At the end of the Introduction, the objectives of the study are clearly stated.
Lines 54 – 57:
This statement needs to be supported with more references on the influence of the many stated factors on the N mineralization rate.

Experimental design

The work has sufficient originality in the sense that previous work on nitrogen mineralization rates on Karst ecosystems at different stages of restoration/'succession' with a comparison to a nearby agricultural soil is scarce. The work also falls within the scope of the journal.

The research question is adequately described. The knowledge gap that this work intends to fill is identified.

However, Materials and Methods suffers from crucial omission of essential information, especially on soil sampling methodology and statistical analysis.

In particular, more details are required on how the sampling point/s was/were selected for it to be representative of the prevailing spatial variation of soil properties. Importantly, according to the description given, it appears that only one sampling point was used in each ecosystem. If this is so, then there is a serious question over the validity of the results if the selected sampling point cannot be demonstrated as representative of the whole ecosystem. Given the usually high spatial heterogeneity in soil physical and chemical properties in almost all natural ecosystems, it is important that soil samples are representative of the existing heterogeneity. Therefore, soil sampling from an adequate number of replicate sampling points is essential. In lines 132-133 of the text, it is mentioned that sampling was done in triplicate from each sampling point, which appears to say that there was only one sampling point per ecosystem with three replicate samples at the selected sampling point. The data files that have been supplied contain labels for the measured soil variables (i.e. columns), but do not provide adequately labelled information about the ecosystems, sampling points and replicates (i.e. rows).

There is a major flaw in the statistical analysis. One-way ANOVA is not appropriate for statistical analysis of this study. There are two factors, namely, ecosystem and soil depth. Therefore, the data should be analyzed as a two-factor factorial (i.e. two-way ANOVA).

Furthermore, details about the variables that were used in the Mantel’s test should be stated along with the rationale for their selection.

Further comments and queries are given as annotations on the manuscript.

Hence, there are serious issues with the technical standard of the work.

A wide range of soil properties have been measured, apparently, using correct analytical methodology. However, inadequacy of replication during sampling appears to reduce the validity of these measurements, especially when they are subjected to statistical analysis and when they are used to draw conclusions.

Validity of the findings

The description of results does not contain information on statistical significance and therefore, apparently, is based on numerical values without any consideration of their dispersion. Consequently, it is not possible to assess the Discussion and Conclusions in their present form. As such, there are serious questions on the validity of the findings as reported in the present form of the manuscript. Specific comments on this issue are given below:

There are major flaws in the presentation of results. These are:

The results appear to be based on one-way ANOVA. As stated above in my comments on materials and methods, two-way ANOVA should be used for statistical analysis to determine the significance of the effects of ecosystem, soil depth and ecosystem × soil depth interaction. Therefore, a description of results based on an incorrect statistical analysis would not be valid.

With the exception of reporting the results of the Mantel’s test, information about statistical significance of the observed variations is not given for any of the measured variables. Significance of the differences between ecosystems for each of the measured variables, significance of their variation with soil depth and the significance of the ecosystem × soil depth interaction effect is not stated. Without providing such information, the differences are described based on their numerical values. This is not scientifically valid. Therefore, I have not evaluated the accuracy of the statements given in the Results section. Correction of this section would mean a complete re-writing of it.

The figures do not meet the quality standards of a manuscript. In the version that is available for me to view, the axis labels appear in Chinese script. The figure caption is not sufficiently descriptive. Even though there are error bars, it is not specified what they are. Are they standard deviations, standard errors of means or confidence intervals of means?

As stated earlier with regard to experimental design, there are questions over the data in terms of the validity of the sampling scheme and the level of replication to ensure that the data are adequately representative of the soil variability within each ecosystem. There is also the issue that the provided data is incomplete with no clear description of the classification variables to enable independent verification of the accuracy of the statistical analysis.

Discussion and Conclusions

Because of the fundamental flaws in the statistical analysis and the method of reporting the results, the Discussion in its present form is not valid. Therefore, I have not assessed the statements given in the Discussion.

The present version of the Discussion appears to be a repetition of results. Instead, the Discussion should seek to explain the observed variation patterns, build inter-relationships between different aspects of the study and provide answers to the research questions posed at the beginning. The Discussion should also compare the results obtained from this work with similar or related work reported in literature. This has been done to a certain extent in the Discussion. However, because of the fundamental flaws in the statistical analysis of data, their interpretation and description in the Results section, a meaningful assessment of the Discussion cannot be done. The same applies to the Conclusions.

Additional comments

If and when a revision is undertaken, the authors should seek to apply a data reduction method such as Principal Component Analysis or Factor Analysis (or any other relevant statistical method) to reduce the number of variables that they use to explain the observation in nitrogen mineralization rates. For example, results of the Mantel's test as shown in Figure 8 of the manuscript shows that there are significant correlations among the many variables that have been measured. In the present manuscript, all these individual values are invoked in trying to explain the results. This does not provide a clear or coherent explanation. Instead, the authors should seek to reduce the data by identifying independent Principal Components or underlying Factors and use them to explain the observed variations. This will enable the authors to provide more insights in to the underlying soil processes which have caused the observed variation in nitrogen mineralization rates during the chronosequence represented by the different ecosystems that have been the focus of this work.

Reviewer 3 ·

Basic reporting

The topic of the paper entitled “Effects of vegetation restoration in karst areas on soil nitrogen mineralisation” is noteworthy and falls within the scope of PeerJ. In this manuscript, the authors have analysed the effect of vegetation of soil nitrogen mineralization using the scientific methods. The research topic and data collected are intrinsically interesting because such types of the studies impact the cause-effect relationships between environment, biodiversity and society. The research topic and findings are inspiring. The suggestions made in the reviewed manuscript (attached here with) may help to improve the quality of the Ms. further.

Experimental design

Appropriately designed

Validity of the findings

Appropriate

Additional comments

The introduction does establish the existing state of knowledge but needs minor revision as suggested in the text attached herewith.
Authors should add the one line hypothesis at the end of the introduction with proper citations.
The citations need up-dated.
English needs minor improvement.
I have made more corrections and suggestions directly in the manuscript.
Please revise the paper accordingly.

Annotated reviews are not available for download in order to protect the identity of reviewers who chose to remain anonymous.

---

## Round 0.2 · Major Revisions

One reviewer raised several important issues in this latest round of review with regards to sampling and statistical methods. The authors must address these concerns.

·

Basic reporting

No Comment apart from those of my previous review of the original manuscript.

Experimental design

Two major flaws in the experimental methodology were highlighted in my previous review.

(1) Lack of clarity on the sampling and whether adequate replication has been done.

Details have been given on the sampling (i.e. sampling done according to a 'S'-patterned scheme). However, this does not clarify the point whether soil samples have been obtained from an adequate number of sampling points. According to the description given, in each ecosystem, soils have been obtained from three sampling points. In my opinion, this number of sampling points is not adequate, especially given the high variability of soil properties, as displayed in all the graphs showing the results.

(2) One-way ANOVA not being the correct method of analysis to determine the influence of ecosystem and soil depth on soil properties. My recommendation was to apply a two-way ANOVA which will enable determination of not only the significance of the effects of ecosystem, but also the effects of soil depth and the ecosystem × soil depth interaction.

The authors have decided to remain with the one-way ANOVA, arguing that: (a) Ecosystem differences were their main focus; (b) Considering the effect of soil depth in the analysis did not alter the ecosystem effect; (c) The ecosystem × soil depth interaction effect was not significant.

However, I am not convinced that these justifications are sufficient to stay with the one-way ANOVA because of the following reasons (referring to the justifications given above):

(a) I agree that the ecosystem effect is the main focus, but one needs to separate the soil depth effect to determine whether the ecosystem effect is significant or not. In a one-way ANOVA, the soil depth effect is in the error term.

(b) Soil depth effect is not significant because of the very high variability of many soil properties among the three sampling points, which also shows that the number of sampling points is not adequate. This can be seen by the very wide error bars in all figures. The mean values of several soil properties show that there is variation with soil depth.

(c) Variations of many soil properties as shown in the figures show that different ecosystems showed different variation patterns with soil depth. Again, these are not statistically significant because of the inadequate replication. Furthermore, it can be noted that despite the authors' contention that ecosystem × soil depth interaction was not significant, in their description of results, the authors' describe differences in patterns of variation among ecosystems at different depths. This is an indirect acknowledgement that ecosystem × soil depth interaction is significant and therefore needs to be taken in to account.

Validity of the findings

Description of the findings and their discussion has not changed significantly from the initial version of the manuscript. Therefore, my comments given in the previous review apply to the revised version also, especially because the authors have decided not to change the method of statistical analysis.

Additional comments

The fundamental flaws in the soil sampling, data analysis and interpretation and discussion of results that I had highlighted in my previous review remain in the revised manuscript as well.

Reviewer 3 ·

Basic reporting

The authors have appropriately revised the paper. I am satisfied with the revised version of the paper. Recommended for acceptance.

Experimental design

Appropriate

Validity of the findings

Satisfactory

Additional comments

Be accepted

---

## Round 0.3 · accepted · Accept

The authors have addressed the comments of the reviewer. The reviewer indicates that the authors are unlikely to address concerns raised previously, and that if the other reviewer accepted the paper, it can be accepted.

·

Basic reporting

Please refer to my earlier comments on two rounds of review. They apply to this round of review as well.

Experimental design

Please refer to my earlier comments on two rounds of review. They apply to this round of review as well.

Validity of the findings

Please refer to my earlier comments on two rounds of review. They apply to this round of review as well.